# Mechanical Ventilation Strategies Targeting Different Magnitudes of Collapse and Tidal Recruitment in Porcine Acid Aspiration-Induced Lung Injury

**DOI:** 10.3390/jcm8081250

**Published:** 2019-08-18

**Authors:** Juliane Haase, Dorina C. Buchloh, Sören Hammermüller, Peter Salz, Julia Mrongowius, Nadja C. Carvalho, Alessandro Beda, Anna Rau, Henning Starke, Peter M. Spieth, Claudia Gittel, Thomas Muders, Hermann Wrigge, Andreas W. Reske

**Affiliations:** 1Department of Anesthesiology and Intensive Care Medicine, University Hospital Leipzig, 04103 Leipzig, Germany; 2Department of Anesthesiology, Intensive Care Medicine and Pain Therapy, St. Georg Hospital, 04129 Leipzig, Germany; 3Department of Visceral, Transplantation, Vascular and Thoracic Surgery, University Hospital Leipzig, 04103 Leipzig, Germany; 4Innovation Center Computer Assisted Surgery (ICCAS), University of Leipzig, Medical Faculty, 04103 Leipzig, Germany; 5Laboratory of Pneumology LIM09, Medical Faculty, São Paulo University, São Paulo 05508-060, Brazil; 6Department of Electronic Engineering, Federal University of Minas Gerais, Belo Horizonte 31270-901, Brazil; 7Department of Anesthesiology, University Medicine Göttingen, University of Göttingen, 37075 Göttingen, Germany; 8Department of Anesthesiology and Intensive Care Medicine, Hannover Medical School, 30625 Hannover, Germany; 9Department of Anesthesiology and Critical Care Medicine, University Hospital Dresden, Technische Universität Dresden, 01307 Dresden, Germany; 10Department for Horses, University of Leipzig, 04103 Leipzig, Germany; 11Department of Anesthesiology and Intensive Care Medicine, University Hospital Bonn, 53127 Bonn, Germany; 12Department of Anesthesiology, Intensive Care and Emergency Medicine, Pain Therapy, Bergmannstrost Hospital Halle, 06112 Halle, Germany; 13Department of Anesthesiology, Intensive Care Medicine, Emergency Medicine and Pain Therapy Heinrich-Braun-Hospital Zwickau, 08060 Zwickau, Germany

**Keywords:** acute respiratory distress syndrome, lung protective mechanical ventilation, positive end-expiratory pressure, lung recruitment, electrical impedance tomography

## Abstract

Reducing ventilator-associated lung injury by individualized mechanical ventilation (MV) in patients with Acute Respiratory Distress Syndrome (ARDS) remains a matter of research. We randomly assigned 27 pigs with acid aspiration-induced ARDS to three different MV protocols for 24 h, targeting different magnitudes of collapse and tidal recruitment (collapse&TR): the ARDS-network (ARDSnet) group with low positive end-expiratory pressure (PEEP) protocol (permissive collapse&TR); the Open Lung Concept (OLC) group, PaO_2_/FiO_2_ >400 mmHg, indicating collapse&TR <10%; and the minimized collapse&TR monitored by Electrical Impedance Tomography (EIT) group, standard deviation of regional ventilation delay, SD_RVD_. We analyzed cardiorespiratory parameters, computed tomography (CT), EIT, and post-mortem histology. Mean PEEP over post-randomization measurements was significantly lower in the ARDSnet group at 6.8 ± 1.0 cmH_2_O compared to the EIT (21.1 ± 2.6 cmH_2_O) and OLC (18.7 ± 3.2 cmH_2_O) groups (general linear model (GLM) *p* < 0.001). Collapse&TR and SD_RVD_, averaged over all post-randomization measurements, were significantly lower in the EIT and OLC groups than in the ARDSnet group (collapse *p* < 0.001, TR *p* = 0.006, SD_RVD_
*p* < 0.004). Global histological diffuse alveolar damage (DAD) scores in the ARDSnet group (10.1 ± 4.3) exceeded those in the EIT (8.4 ± 3.7) and OLC groups (6.3 ± 3.3) (*p* = 0.16). Sub-scores for edema and inflammation differed significantly (ANOVA *p* < 0.05). In a clinically realistic model of early ARDS with recruitable and nonrecruitable collapse, mechanical ventilation involving recruitment and high-PEEP reduced collapse&TR and resulted in improved hemodynamic and physiological conditions with a tendency to reduced histologic lung damage.

## 1. Introduction

In patients with Acute Respiratory Distress Syndrome (ARDS) mechanical ventilation with low tidal volumes (V_T_) can reduce complications such as ventilator associated lung injury (VALI) and mortality [1,2,3]. It remains uncertain, however, which magnitude of collapse and associated tidal recruitment (TR, i.e., cyclic opening and re-collapse during mechanical ventilation) should be targeted: while individualized high positive end-expiratory pressure (PEEP) and/or lung recruitment seems to be beneficial in some patients by reducing TR, it can instead be harmful in others [4,5,6,7].

A pragmatic approach individually selecting the PEEP level based on the ARDS severity is the low-PEEP table of the ARDS-network (ARDSnet), combining low V_T_ ventilation and PEEP-FiO_2_ combinations depending on oxygenation [2,8]. Among the advantages of the ARDSnet PEEP/FiO_2_-table are its straightforward use and the limitation of V_T_ and plateau airway pressure potentially reducing VALI.

Another approach to reduce VALI is based mainly on physiological rationale and involves reducing collapse and TR (collapse&TR) by combining lung re-aeration by recruitment maneuvers (RM) and subsequent decremental PEEP-titration along the expiratory limb of the individual pressure–volume curves [5,9,10,11,12,13,14]. Many parameters (e.g., oxygenation, compliance, quantitative computed tomography) can be estimated during a decremental PEEP-titration and can be used to identify an “individually optimized” lung condition and the corresponding PEEP to maintain it [9,10,15,16,17,18]. However, if lung collapse was assessed again after commencing mechanical ventilation with the “individualized” PEEP, it was seldom truly minimized or zero, suggesting the presence of nonrecruitable lung tissue [9,10,19].

Electrical Impedance Tomography (EIT) is a non-invasive bedside method for functional lung imaging. Parameters derived from EIT have been proposed to aid in detecting TR and/or collapse [20,21,22,23]. For example, the EIT-derived regional ventilation delay (RVD) and more specifically its variability considering all pixels (standard deviation of regional ventilation delay, SD_RVD_), was proposed for selecting individualized PEEPs high enough to minimize TR but also low enough to avoid PEEP-related complications (e.g., hyperinflation) [23]. Such attractive features of the SD_RVD_-approach and especially the avoidance of over-recruitment await further support and comparison with other approaches to mechanical ventilation [23].

In this study we used the clinically realistic and anesthesia-relevant acid-aspiration model for pulmonary (direct) ARDS, which so far has been deemed less recruitable. We performed a randomized comparison of three groups of pigs, which received mechanical ventilation for 24 h with PEEP individualized as follows: ARDSnet low-PEEP table, EIT-derived SD_RVD_, or Open-Lung-Concept (OLC) [2,9,23]. We analyzed changes in EIT, computed tomography (CT), histology, and cardiorespiratory parameters. We hypothesized that ventilation with PEEP individualized according to the EIT-derived SD_RVD_ reduces collapse&TR to an extent comparable with the OLC and is associated with less histological damage than ARDSnet.

## 2. Materials and Methods

### 2.1. Animal Handling and Preparation

After approval by the animal ethics committee (Regierungspräsidium Leipzig, Leipzig, Germany, reference number TVV35/11, date of approval: 22 July 2008), animal experiments complying with the ARRIVE guidelines were conducted in the Surgical Clinic, Faculty of Veterinary Medicine, University of Leipzig, Leipzig, Germany [24]. Figure 1 illustrates the study flow-chart. Appendix A illustrates the sequence of measurements.

Pigs (weight 36.8 ± 5.0 kg) were fasted and premedicated with midazolam (1 mg·kg^−1^ intramuscular) and ketamine (15 mg·kg^−1^ intramuscular). Animals were anaesthetized by infusion of fentanyl 5 (3–30) mg·kg^−1^·h^−1^, midazolam 2 (1–6) mg·kg^−1^·h^−1^, ketamine 15 (5–30) mg·kg^−1^·h^−1^, and pancuronium 0.15 mg·kg^−1^·h^−1^ and remained on the CT-table in supine position throughout the study.

After tracheostomy, baseline (BL) ventilation was initiated (PEEP 5 cmH_2_O, V_T_ 6 mL·kg^−1^ bodyweight, inspiratory-to-expiratory (I:E) ratio 1:2, FiO_2_ 30%). Respiratory rate (RR) was adjusted to maintain pH > 7.30. Sterile surgical instrumentation included arterial, central venous, pulmonary artery, and suprapubic urine catheters. Heparin (200 IE·h^−1^), Ringer-Acetat (5 mL·kg^−1^·h^−1^) and Cefuroxime (prophylaxis, 750 mg every 6 h) infusions were started after instrumentation.

Mean arterial pressure (MAP), mean pulmonary artery pressure, central venous pressure, heart rate (HR) and body temperature were monitored and waveforms were recorded continuously. Cardiac output (CO) was measured by transpulmonary thermodilution and systemic vascular resistance (SVR) and pulmonary vascular resistance (PVR) were calculated. Airway flow and pressure waveform signals were acquired continuously from the ventilator. Peak airway pressure (P_aw-peak_), plateau airway pressure (P_aw-plateau_), respiratory system compliance (compliance), and driving pressure (=P_aw-plateau_ minus PEEP) were calculated using standard procedures [14]. P_aw-plateau_ and PEEP were measured during zero flow conditions. All blood gas measurements (arterial and mixed-venous, co-oximetry) were performed after 5 min of ventilation with pure oxygen (100% O_2_). Intrapulmonary shunt (shunt) was calculated [25].

After the study, animals were euthanized by injecting 50 mmol potassium chloride during deepened anesthesia.

### 2.2. Acid Aspiration-Induced Pulmonary Acute Respiratory Distress Syndrome

Pulmonary ARDS was induced by intra-tracheal hydrochloric acid instillation (HCl, 0.1 molar). After five minutes of 100% O_2_-ventilation, a suctioning catheter was advanced through the endotracheal tube and 60 mL of HCl were instilled. Lung injury compatible with moderate ARDS was considered established when the PaO_2_ (100% O_2_) remained lower than 200 mmHg for 30 min under BL ventilation [26]. If this criterion was not reached, additional 30 mL HCl aliquots were instilled until PaO_2_ remained stable below 200 mmHg for 30 min.

### 2.3. Individualization of PEEP and Mechanical Ventilation

After establishing ARDS, 24 pigs were randomized (sealed envelopes) to one of three ventilation protocols using different individualized PEEPs for 24 h: ARDSnet [2], OLC [9], and EIT-derived SD_RVD_ (EIT) [23].

In the OLC and EIT groups, PEEP was titrated according to different surrogates detecting the lung condition considered “optimal” according to the concept (see below). All decremental PEEP titrations were performed during volume-controlled ventilation (VCV, V_T_ = 6 mL·kg^−1^ bodyweight) with 100% oxygen. An RM lasting two minutes during pressure-controlled ventilation (inspiratory pressure = 60 cmH_2_O, PEEP = 40 cmH_2_O) preceded decremental PEEP titrations. After recruitment, PEEP was reduced to 30 cmH_2_O (or 26 cmH_2_O in the case of intractable hemodynamic instability) and then decreased in steps of 2 cmH_2_O until either a PEEP = 6 cmH_2_O was reached or the SaO_2_ decreased below 90%. Every PEEP-level required about 10 min to allow for all necessary measurements (i.e., CT, EIT). In OLC animals, the PEEP-step, at which a 10% reduction of the individual maximum oxygenation (PaO_2_ during 100% O_2_ ventilation) occurred, was identified and 2 cmH_2_O added to obtain the PEEP used for subsequent ventilation [9]. In EIT animals, a slow inflation (inspiratory flow 4 L·min^−1^, inflation volume 12 mL·kg^−1^) was performed at every PEEP step and the minimal SD_RVD_ identified the PEEP for subsequent ventilation (see below) [23]. RM and PEEP titration were repeated during the experiment based on objective experience-based criteria indicating derecruitment: an oxygenation decrease below 400 mmHg for OLC and a 20% increase of the SD_RVD_ from the previous measurement point for EIT [10,27].

### 2.4. Quantitative Computed Tomography (qCT)

Whole-lung CT scans were taken without contrast medium. After manual segmentation of the lung images, qCT-parameters were calculated using standard procedures. Total lung mass (M_total_) and volume (V_total_) were identified within [−1000 to 100] Hounsfield units (HU). Differently aerated lung compartments were defined by the following intervals and calculated as a percentage of M_total_: hyperinflated (M_hyper_, [−1000 to −901] HU), normally aerated (M_normal_, [−900 to −501] HU), poorly aerated (M_poor_, [−500 to −101] HU), and non-aerated (M_non_, [−100 to 100] HU) [27,28,29,30]. We calculated TR by subtracting inspiratory from expiratory M_non_ [9]. To characterize the time-weighted cumulative impact that different amounts of TR had on the lung parenchyma, we conceived the surrogate ‘tidal-recruitment-hours’ (TR-hours); we multiplied the amount of TR at a given measurement point by the period it had acted since the last measurement. These products from all measurement points were then summed up to give TR-hours.

### 2.5. Electrical Impedance Tomography

The PulmoVista 500 EIT-system (Dräger Medical Germany GmbH, Lübeck, Germany) was used. The SD_RVD_, representing inhomogeneous regional ventilation (EIT-surrogate for TR), was calculated offline from the EIT-data sampled during slow inflation [23]. SD_RVD_-hours were calculated in analogy to TR-hours. The regional distribution of ventilation was described using the center of ventilation (CoV) [31,32] with reconstructed tidal images based on Dräger algorithms (see Appendix A).

### 2.6. Tissue Processing and Histological Analysis

Four tissue samples (ventral, medial-ventral, medial-dorsal, and dorsal) were taken from the left lower lobe. Tissue samples were fixed per standardized protocol and stained with hematoxylin-eosin (see Appendix A). We assessed the criteria hemorrhage, inflammation, and intra-alveolar edema by light microscopy and calculated the cumulative Diffuse Alveolar Damage (DAD) Score as the sum over all criteria [33,34]. The wet-to-dry ratio of the lung was measured as well.

### 2.7. Statistical Analysis

Our sample size estimation indicated that eight pigs per group were required (see Appendix A). Data are presented as mean and standard deviation (SD) or 95% confidence interval (95% CI). Changes from BL to established ARDS (randomization) were analyzed using paired t-tests. General Linear Model (GLM) approaches were used to analyze changes between ARDS and 24 h for group, time, and interactions effects. Only if the GLM indicated significant differences, Sidak’s post-hoc tests were used. Correlation between continuous variables was derived from linear regression. We considered *p* values <0.05 significant.

## 3. Results

### 3.1. General Aspects and Effects of Induction of ARDS

To study 24 animals for a per-protocol analysis, we had to enroll 27 pigs. The mean amount of HCl required for inducing moderate ARDS (ARDSnet 124 ± 20, OLC 105 ± 30, EIT 135 ± 16 mL) differed marginally between groups (GLM *p* = 0.046, post-hoc: OLC vs. EIT *p* = 0.045). Induction of ARDS decreased compliance and PaO_2_ while PaCO_2_ and shunt increased (all *p* < 0.001, Table 1). Induction of ARDS increased collapse&TR significantly (M_non_ BL 19.8% ± 11.8%, ARDS 51.8% ± 8.0%, *p* < 0.001, TR BL 6.1% ± 4.5%, ARDS 15.8% ± 12.0%, *p* = 0.001) and also caused significant changes in, M_normal_, M_poor_, gas content, M_total_, and V_total_ as well as in the SD_RVD_ (all *p* ≤ 0.01, Figure 2, Appendix A).

According to EIT, CoV did not change after induction of ARDS (*p* = 0.30, BL 55.8% ± 2.5%, ARDS 56.5% ± 3.1% *p* = 0.30, Appendix A). We noticed randomization bias for PaO_2_ after ARDS induction (GLM *p* = 0.016, posthoc: ARDSnet vs. OLC *p* = 0.022, ARDSnet vs. EIT *p* = 0.063, OLC vs. EIT *p* = 0.95, Table 1) as well as for shunt (GLM *p* = 0.005, posthoc: ARDSnet vs. OLC *p* = 0.004, ARDSnet vs. EIT *p* = 0.11, OLC vs. EIT *p* = 0.40), but not for compliance or M_non_. Compared to BL, induction of ARDS decreased HR (*p* = 0.003) and increased SVR (*p* = 0.02) and PVR (*p* < 0.001), while CO and MAP did not change significantly (Table 1).

### 3.2. Effects on Lung Morphology and Function

Each group developed characteristic lung conditions with significant between-group differences (Table 1, Figure 2, Appendix A).

Mean PEEP over all post-randomization measurements was significantly lower (GLM *p* < 0.001) in ARDSnet (6.8 ± 1.0 cmH_2_O) compared to EIT (21.1 ± 2.6 cmH_2_O) and OLC (18.7 ± 3.2 cmH_2_O) (both *p* < 0.001), but not between EIT and OLC (*p* = 0.17). Mirroring the differences in PEEP, ARDSnet developed significantly more collapse&TR than OLC and EIT (M_non_ and TR in Figure 2, Appendix A). Neither collapse nor TR differed significantly between OLC and EIT (both *p* = 1.0). OLC and EIT groups showed reduced M_non_ (*p* < 0.001), TR (*p* = 0.006), and shunt (*p* < 0.001), and increased compliance (*p* < 0.001) and PaO2 (*p* < 0.001) in OLC and EIT groups. Gas content, V_total_, M_normal_, and M_poor_ changed significantly over time and between groups, with significant interaction (Appendix A, all *p* ≤ 0.01). In ARDSnet gas content, V_total_, M_normal_ (all *p* ≤ 0.001), and M_poor_ (*p* < 0.01) were significantly smaller than in OLC and EIT and again without differences between EIT and OLC (*p* ≥ 0.3). The EIT-derived CoV indicated that ventilation occurred in more ventral regions in ARDSnet compared to OLC and EIT, in which recruitment and individualized PEEP had shifted ventilation more dorsally (Appendix A).

Significantly more TR occurred in ARDSnet (362 ± 152 TR-hours) compared to OLC (64 ± 57 TR-hours) and EIT (67 ± 38 TR-hours) (GLM *p* < 0.001, post-hoc: ARDSnet vs. OLC or EIT *p* < 0.001, EIT vs. OLC *p* = 1.0). Although not directly comparable, SD_RVD_-hours showed a compatible behavior (ARDSnet 211 ± 53 SD_RVD_-hours, OLC 120 ± 28 SD_RVD_-hours, EIT 105 ± 42 SD_RVD_-hours) (GLM *p* = 0.001, post-hoc: ARDSnet vs. OLC or EIT *p* = 0.002, EIT vs. OLC *p* = 1.0, Appendix A). TR quantified by qCT correlated strongly with EIT-derived SD_RVD_, as did the time-weighted parameters TR-hours and SD_RVD_-hours (R^2^ values 0.66 and 0.7, respectively, *p* < 0.001, Appendix A).

SD_RVD_ increased until 16 h and decreased subsequently in ARDSnet, whereas in OLC and EIT it decreased after recruitment and remained significantly lower than in ARDSnet (GLM time effect *p* = 0.004, group effect *p* = 0.002, interaction effect *p* = 0.002, post-hoc: ARDSnet vs. OLC or EIT *p* < 0.03, EIT vs. OLC *p* = 0.83, Figure 2 and Appendix A).

Compliance was significantly higher in OLC and EIT than ARDSnet (GLM, group effect *p* < 0.001, interaction effect *p* = 0.002, and post-hoc all *p* < 0.01, Appendix A), but did not differ between OLC and EIT (*p* = 0.76).

In ARDSnet, PaO_2_ was lower and shunt was higher than in OLC and EIT groups (*p* ≤ 0.001 for all time points, group and post-hoc tests), but no difference was found between OLC and EIT (*p* = 0.91 for shunt *p* = 0.84 and for PaO_2_). The PaCO_2_ did not vary significantly between groups (*p* = 0.1) nor over time (*p* = 0.17).

### 3.3. Hemodynamics and Systemic Physiological Parameters

Most hemodynamic parameters remained largely unchanged for the first 8 h of ventilation. Afterwards, a systemic inflammatory response pattern developed progressively until 24 h. These results corroborate with the significant changes over time of SVR, PVR, MAP, HR, and body temperature (all *p* ≤ 0.001, without any between-group effects or interaction, Table 1 and Appendix A). CO neither changed significantly with time nor between groups (all *p* ≥ 0.13). At 24 h, the cumulative fluid balance differed significantly between groups (ARDSnet 111 ± 64, OLC 131 ± 69, EIT 211 ± 45 mL/kg bodyweight, ANOVA at 24 h: *p* = 0.008, post-hoc: ARDSnet vs. OLC *p* = 0.88, ARDSnet vs. EIT *p* = 0.01, OLC vs. EIT *p* = 0.045, Appendix A).

### 3.4. Assessment of Edema and Histology

During individualized ventilation, M_total_ (as a surrogate of edema) increased significantly until 8 h and remained elevated afterwards (time *p* < 0.001), without significant between-group differences (*p* = 0.12), or interaction (*p* = 0.06). The wet-to-dry ratio differed significantly between groups (ARDSnet 8.1 ± 0.4, OLC 7.7 ± 0.6, EIT 7.4 ± 0.4; GLM *p* = 0.02; post-hoc: ARDSnet vs. OLC *p* = 0.29, ARDSnet vs. EIT *p* = 0.02, OLC vs. EIT *p* = 0.46).

Figure 3 shows representative macroscopic and related microscopic specimens and summarizes the histologic analyses.

Because regional histological analysis detected no meaningful or significant differences (Appendix A), we averaged regional scores into global scores for the cumulative DAD as well as for each histological criterion [34]. Cumulative DAD scores were 10.1 ± 4.3, 6.3 ± 3.3, and 8.4 ± 3.7 for ARDSnet, OLC, and EIT, respectively (GLM *p* = 0.16). Intra-alveolar edema scores were lowest for OLC (1.8 ± 1.0), followed by EIT (2.1 ± 1.4) and ARDSnet (3.6 ± 1.7) (GLM *p* = 0.042, ARDSnet vs. OLC *p* = 0.058, ARDSnet vs. EIT *p* = 0.13, OLC vs. EIT *p* = 0.98). Inflammation showed similar results (OLC 2.7 ± 1.2, EIT 3.8 ± 1.1, ARDSnet 4.3 ± 1.2, *p* = 0.034, post-hoc: ARDSnet vs. OLC *p* = 0.037, ARDSnet vs. EIT: *p* = 0.85, OLC vs. EIT *p* = 0.17). Hemorrhage did not significantly differ between groups (OLC 1.8 ± 1.4, EIT 2.5 ± 1.4, ARDSnet 2.2 ± 1.7, *p* = 0.63). Comparison of cumulative DAD-scores with TR-hours showed a similar tendency towards higher scores and more cumulative TR in the ARDSnet group for edema and inflammation (Appendix A).

## 4. Discussion

Based on imaging and physiological analyses obtained over an extended experimental period of 24 h, our study adds evidence that some concepts for mechanical ventilation involving individually-titrated PEEP are capable of reducing collapse&TR and restoring lung function even in early acid aspiration-induced ARDS, often considered poorly recruitable. Our results also support the idea that PEEP individualized according to different physiology-based approaches significantly exceeds the PEEP suggested by the ARDSnet concept. EIT and OLC approaches identified similarly high PEEP-values. Physiological and morphological measurements showed improved aeration and ventilation for OLC and EIT as compared with ARDSnet. These differences involved the vast majority of surrogate parameters studied, including key markers for VALI such as driving pressure and TR. Histology revealed a shift to more edema and inflammation in the EIT group than in the OLC group, presumably due to overuse of recruitment. A tendency to higher edema scores were observed in the ARDSnet group despite the fact that per protocol fluid intake was higher in the OLC and EIT groups.

The identification of patients with a high likelihood of a positive response to and potential benefit from lung recruitment and PEEP has been an ongoing quest [4,9], and recent studies are controversial as they either challenge [5] or support [35] this concept. In patients with limited potential for lung recruitment, recruitment and high PEEP may even be detrimental [5].

Our results add further indications that the very early stages of ARDS, even in ARDS forms previously considered poorly recruitable such as acid-aspiration induced ARDS, may offer a window of time during which recruitment and individualized, physiology-based selection of high PEEP may help to restore lung function and to avoid potentially injurious forms of mechanical ventilation. In our study, EIT, qCT, oxygenation, and compliance agreed in demonstrating the improvement in lung aeration and function after recruitment. Excluding qCT from this list because of unwarranted radiation exposure, the pattern of EIT, oxygenation, and/or compliance changes could easily help to answer whether a patient responds to recruitment by improved lung aeration and function. Positive responses to recruitment and PEEP are more likely during (or even limited to) the early phase of DAD [36]. In contrast to patients developing ARDS from bacterial pneumonia, who often have significant non-recruitable lung consolidation and frequently reach the intensive care unit (ICU) several days after the onset of DAD, patients developing lung dysfunction and respiratory distress after aspiration or trauma are in the very early stages of DAD [37,38]. Using EIT, oxygenation, and/or compliance for individualization of PEEP would be completely different from a “recruitment is good for all ARDS patients” approach, which was tested in previous studies [5,39] with negative results. Recent data further support this notion; there are distinct subgroups within the syndrome ARDS to which therapeutic bundles should be individually tailored [40,41,42].

Besides confirming that EIT can aid in detecting a response to recruitment by visualizing recruitment-induced restoration of ventilation in previously non-ventilated regions of the EIT-image (Figure 4, Appendix A), our results suggest that EIT is capable of detecting relevant TR. While lung collapse that persists during mechanical ventilation does not necessarily lead to VALI, cyclic recruitment and derecruitment of airspaces (tidal recruitment) is deemed a main pathomechanism of VALI [3,43].

It appears important that the large SD_RVD_ detected in non-recruited lungs (i.e., ARDSnet group and measurements before recruitment in OLC and EIT groups) were almost always associated with large qCT-measurements of TR (Appendix A). The time-weighted surrogate SD_RVD_-hours correlated strongly with the qCT-counterpart (Appendix A), and there was also an association with histology (Appendix A). Considering that estimation of TR by repetitive qCT will likely remain clinically impracticable, the EIT-derived SD_RVD_ may be a valuable option to detect TR and confirm recruitability in states of significant lung collapse in early ARDS, demonstrated here for acid aspiration.

The ARDSnet ventilation group in our study showed significantly more collapse in qCT, lower compliance, and more time-weighted TR in qCT and EIT than those of both other groups in which recruitment and individualized high PEEP resulted in restored lung volumes. We refrain here from discussing the pros and cons of the “permissive atelectasis” and “open lung” concepts, as there are abundant excellent reviews on that matter [44,45].

Besides the observations that all OLC and EIT pigs responded to recruitment by an increase in functional lung volume and that both oxygenation and EIT clearly detected this response to recruitment, some differences between otherwise widely similar OLC and EIT results warrant discussion. OLC showed a tendency to lower inflammation and edema scores (Figure 3, *p* < 0.05) although not statistically significant in post-hoc analysis, presumably due to a small sample size.

The difference in the number of re-RM could be an explanation. When there is a lot of recruitable lung, the SD_RVD_ will give a strong signal when this amount of lung parenchyma undergoes TR (see above). On the other hand, if a small SD_RVD_ indicates that all or most recruitable lung tissue is already recruited (RM) and subsequently kept recruited by PEEP, a 20% change in the small SD_RVD_ will be easily reached. We used an arbitrary 20% change in SD_RVD_ to prompt re-recruitment and this obviously led to overutilization of recruitment maneuvers. There were no more beneficial effects of re-RM in terms of PaO_2_/FiO_2_, shunt, or compliance in the EIT-group beyond that reached after the initial recruitment and PEEP-titration at the beginning of the study. The greater number of RM and PEEP titrations performed in the EIT-pigs resulted in larger infusion volumes (per protocol linked to number of RM) and a higher number of inflations and deflations, which may all have contributed to higher DAD scores in the EIT group compared to the OLC group [46,47,48]. Therefore, the maintenance of minimal collapse&TR over 24 h required the repeated use of recruitment and PEEP-titrations, whose side-effects (i.e., volutrauma/barotrauma) seemed to neutralize some benefits (i.e., reduced atelectrauma) in the EIT group.

The ability of the SD_RVD_ to provide physiologically sound and clinically useful information may thus depend on the amount of lung collapse. Small absolute SD_RVD_ or small changes of SD_RVD_ during ventilation at high PEEP levels may be less reliable because of an unfavorable signal-to-noise ratio.

## 5. Limitations

Different amounts of HCL required for reaching physiological ARDS criteria must be considered when interpreting differences found in histology between OLC and EIT groups. The pressure limitation for the low-flow-maneuver was set to 80 cmH_2_O. Pigs with very low compliance may thus have been exposed to very high airway pressures during the low-flow-maneuver. Such values were anecdotally reported for RM [37,49], but should of course be avoided for tidal ventilation. As a matter of fact, direct translation of the results of this porcine study into clinical practice is impossible. Our model of very early ARDS has similarities with the problem of aspiration during induction of anesthesia, however, results cannot be simply extrapolated to human and/or later-stage ARDS. As a result of applying the 3R principle, sample size and power of statistical tests needed to be balanced [50]. In our work, some significances, especially the post-hoc tests for histological analyses (group size of eight animals) may have been affected. Further study seems warranted to improve the EIT-approach for bedside detection of clinically relevant collapse&TR and for PEEP-selection. Dedicated low-flow-maneuvers involving large inflation-volume airway pressures should be replaced by breath-by-breath approaches during ongoing ventilation.

## 6. Conclusions

Overall, recruitment and subsequent individualization of PEEP using the entirely noninvasive EIT-method resulted in lung conditions matching those seen in the oxygenation-focused OLC. In our model of ARDS, both, EIT— and OLC-based PEEP-selection resulted in physiological and morphological conditions that seem to meet contemporary criteria for lung-protective mechanical ventilation. The ARDSnet strategy led to progressive lung de-recruitment and histological results showed a tendency towards more lung inflammation in the ARDSnet group. Including EIT-derived information into patterns of physiological parameters may allow identification of patients who will respond to recruitment by reaeration of previously non-ventilated lung regions and reduction of TR. While EIT-based detection of significant TR and EIT-based PEEP-selection performed comparably to the OLC approach, the ability of EIT to detect the necessity for re-recruitment and re-titration of PEEP remains to be studied. An over-utilization of recruitment and PEEP, possibly (mis)guided by the aim at minimization of parameters only indirectly characterizing lung aeration, may indeed be stretching things too far.

## Figures and Tables

**Figure 1 jcm-08-01250-f001:**
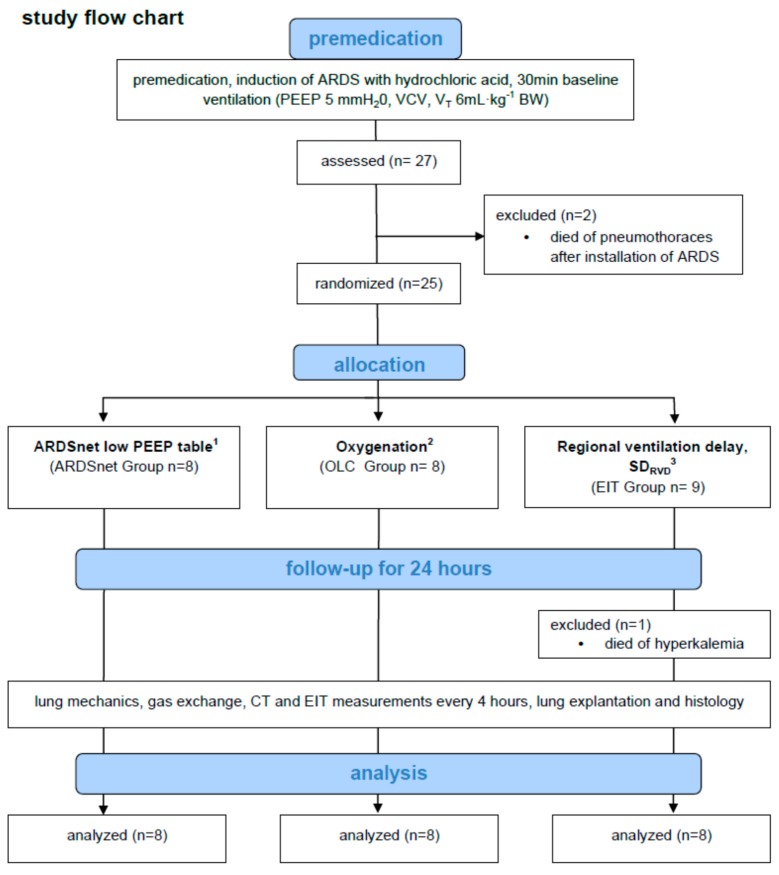
Study flow chart. Pigs were randomized after induction of Acute Respiratory Distress Syndrome (ARDS) and baseline ventilation with positive end-expiratory pressure (PEEP) of 5 mm H_2_0, volume-controlled ventilation (VCV) and tidal volume of 6 mL/kg bodyweight (BW) to one of three ventilation groups referred to in ^1^ [2] for ARDSnet low PEEP table, ^2^ [9] for Open Lung Concept (OLC) and ^3^ [23] for Electrical Impedance Tomography (EIT) with standard deviation of regional ventilation delay (SD_RVD_) measurements. During the follow-up for 24 h computed tomography (CT) and EIT measurements were provided every 4 h. In the EIT group, one pig had to be excluded from the experiment because of intractable metabolic acidosis, thus, this experiment had to be repeated. As a result, 25 pigs were randomized. Because two pigs developed massive pneumothoraces after intra-tracheal hydrochloric acid instillation, we had to enroll 27 pigs in total (see Appendix A).

**Figure 2 jcm-08-01250-f002:**
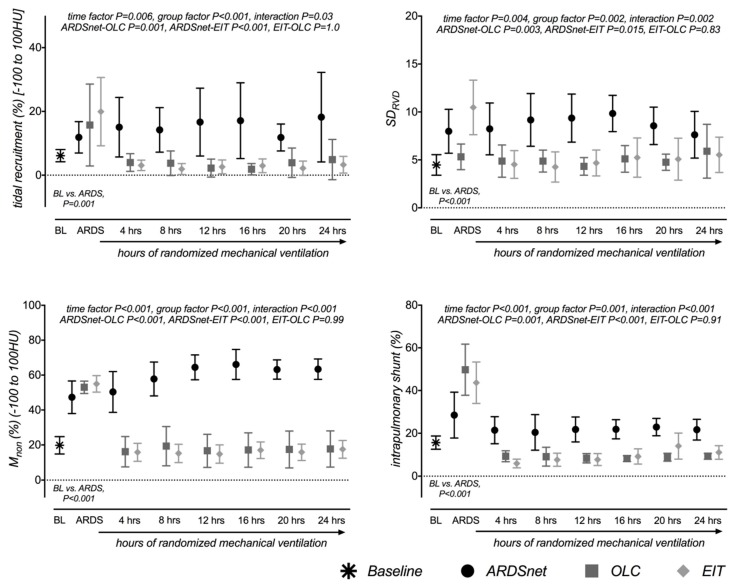
Surrogates of tidal recruitment and lung collapse. Measurements were performed at baseline (BL), ARDS/randomization (ARDS) and every 4 h for 24 h for the three ventilation groups ARDSnet, Open Lung Concept (OLC) and Electrical Impedance Tomography (EIT). Percentage of nonaerated lung tissue mass (M_non_), tidal recruitment (TR), and intrapulmonary shunt are shown in percent. Temporal changes in the standard deviation of regional ventilation delay (SD_RVD_) are shown as surrogates of inhomogeneous lung ventilation. Data are shown as means and their 95% confidence interval. Significant increases, especially in intrapulmonary shunt and M_non_, were shown between BL and ARDS for all parameters (paired t-test). Time and group effects and their interaction were tested using the general linear model approach. Differences between ventilation groups were tested by Sidak’s post-hoc test.

**Figure 3 jcm-08-01250-f003:**
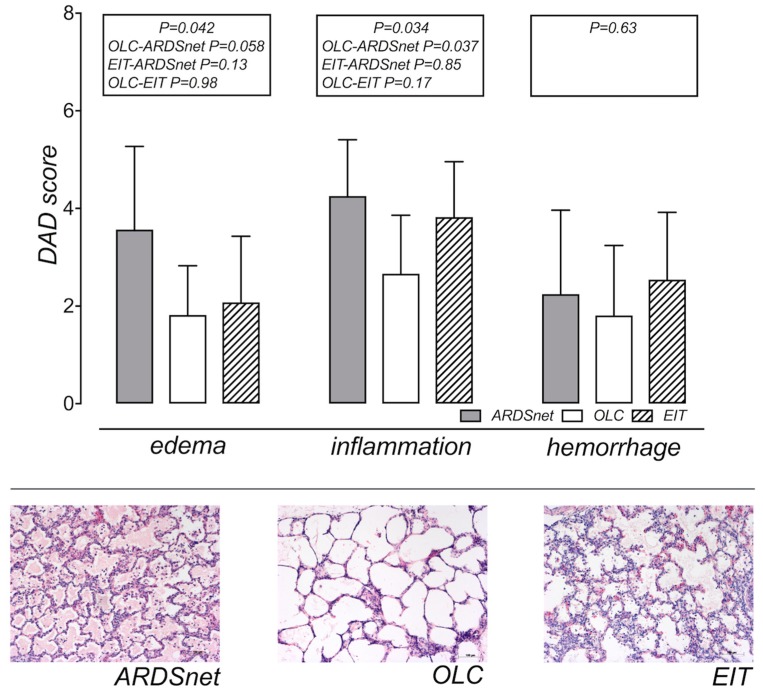
Histological results: diffuse alveolar damage score and histologic samples. The diffuse alveolar damage (DAD) sub-scores (0–12 with increasing severity) are shown for the criteria intra-alveolar edema, inflammation, and hemorrhage as grand mean over all regions for each ventilation group. The bar plots indicate means and standard deviations. Parametric testing was considered appropriate and was done with ANOVA, being significant for inflammation and edema (*p* < 0.05). In Sidak’s post-hoc test OLC vs. ARDSnet differed significantly for inflammation (*p* = 0.037) with a tendency to lower scores for edema (*p* = 0.058). Hemorrhage did not differ between groups (*p* = 0.63). Representative histology samples for the ARDSnet, OLC, and EIT group are shown in the lower panel (100x magnification).

**Figure 4 jcm-08-01250-f004:**
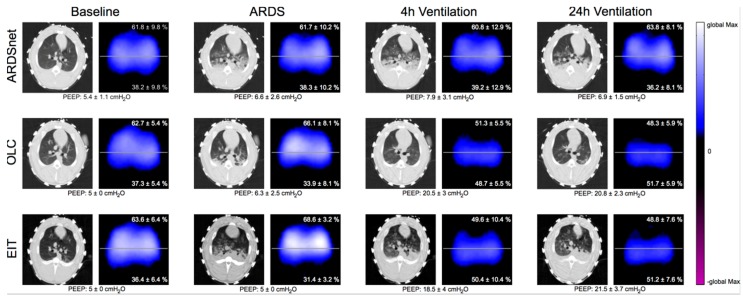
Representative examples of CT- and EIT-images. A derived analysis obtained at baseline, at diagnosis of ARDS, and after 4 and 24 h of mechanical ventilation is presented for each ventilation group (ARDSnet, OLC, and EIT). Representative CT-Images for each group and time point are displayed. For EIT parameter analysis medibus-data (of PEEP) and reconstructed tidal images with Dräger algorithms (Dräger Medical Germany, GmbH, Lübeck, Germany) were used. Percentages for ventral and dorsal distribution of ventilation (positive pixel values) in EIT images were calculated based on tidal images, which were averaged over each group at each point in time for visualization. PEEP data and percentage of ventral and dorsal ventilation are shown as mean with standard deviation for each group and time point.

**Table 1 jcm-08-01250-t001:** Hemodynamic and respiratory parameters.

	Group	Baseline	ARDS	4 h	8 h	12 h	16 h	20 h	24 h
HR ^aa, bbb^ (min^−1^)	ARDSnet OLC EIT	103 ± 23	80 ± 12	105 ± 15	104 ± 13	106 ± 9	108 ± 16	105 ± 20	114 ± 12
			93 ± 13	113 ± 16	115 ± 13	109 ± 11	110 ± 17	117 ± 10	116 ± 18
			84 ± 15	103 ± 23	110 ± 16	114 ± 15	116 ± 25	115 ± 15	114 ± 23
MAP ^bbb^ (mmHg)	ARDSnet OLC EIT	87 ± 20	82 ± 12	89 ± 13	83 ± 11	81 ± 13	79 ± 13	75 ± 9	70 ± 8
			94 ± 10	76 ± 14	75 ± 8	78 ± 7	73 ± 11	76 ± 12	71 ± 10
			91 ± 17	80 ± 15	75 ± 12	71 ± 5	72 ± 11	70 ± 5	63 ± 11
CO (L·min^−1^)	ARDSnet OLC EIT	5.6 ± 1.6	5.0 ± 1.6	5.8 ± 2.6	5.2 ± 1.7	5.3 ± 0.9	5.4 ± 0.9	5.4 ± 1.1	6.2 ± 0.8
			5.6 ± 1.3	4.3 ± 1.0	4.2 ± 1.1	4.3 ± 0.8	4.6 ± 0.8	4.8 ± 0.6	4.7 ± 0.9
			4.8 ± 1.3	4.0 ± 1.2	4.8 ± 1.1	4.6 ± 1.5	4.6 ± 1.4	4.7 ± 1.2	5.0 ± 1.2
SVR ^a, bbb^ (dyn·s·cm^−5^)	ARDSnet OLC EIT	1186 ± 430	1660 ± 954	1334 ± 524	1370 ± 539	1151 ± 314	1078 ± 162	1041 ± 201	888 ± 212
			1331 ± 338	1510 ± 478	1353 ± 499	1246 ± 289	1185 ± 331	1145 ± 222	1047 ± 154
			1409 ± 227	1604 ± 1046	1206 ± 214	1201 ± 475	1156 ± 491	1123 ± 496	1069 ± 450
PVR ^aaa, bb^ (dyn·s·cm^−5^)	ARDSnet OLC EIT	167 ± 73	391 ± 173	394 ± 157	412 ± 115	350 ± 112	324 ± 120	249 ± 90	268 ± 55
			234 ± 69	323 ± 156	260 ± 140	274 ± 133	270 ± 90	250 ± 92	245 ± 76
			333 ± 128	431 ± 252	352 ± 129	291 ± 131	219 ± 77	240 ± 81	235 ± 112
PEEP ^bbb, ccc, ddd^ (cmH_2_0)	ARDSnet ^eee, fff^ OLC EIT	5.0 ± 0	5.0 ± 0	7.9 ± 3.1	7.1 ± 1.9	6.1 ± 1.6	6.4 ± 2	6.9 ± 1.6	6.5 ± 1.6
			5.0 ± 0	18 ± 3.5	18.0 ± 3.5	19.0 ± 3.2	19.0 ± 3.2	19.0 ± 3.2	19.0 ± 3.2
			5.0 ± 0	19.5 ± 2.1	20.0 ± 3.2	21.8 ± 3.3	21.5 ± 2.6	22.0 ± 3.0	21.8 ± 3.5
Driving ^aaa, bbb^, pressure ^ccc, ddd^ (cmH_2_0)	ARDSnet ^eee, fff^ OLC EIT	10.7 ± 2.1	21.2 ± 2.8	19.1 ± 3.9	19.1 ± 2.6	18.1 ± 3.0	18.9 ± 4.8	18.1 ± 4.3	18.4 ± 3.4
			20.3 ± 2.2	12.6 ± 2.3	12.7 ± 2.4	11.6 ± 2.2	11.2 ± 1.8	11.5 ± 1.6	11.5 ± 1.6
			23.3 ± 1.6	12.6 ± 3.5	11.5 ± 3.0	12.6 ± 3.3	12.3 ± 2.1	12.0 ± 2.2	12.3 ± 3.0
P_aw-plat_ ^aaa, bb, ccc, dd^ (cmH_2_0)	ARDSnet ^fff^ OLC EIT	16.5 ± 2.1	28.1 ± 3.6	28.1 ± 4.0	27.0 ± 3.4	26.3 ± 3.5	26.1 ± 3.2	27.8 ± 3.8	28.0 ± 3.2
			27.4 ± 3.3	32.4 ± 4.6	32.7 ± 3.8	32.0 ± 3.9	31.6 ± 4.0	31.1 ± 3.6	31.9 ± 3.7
			30.8 ± 3.3	33.6 ± 3.5	34.8 ± 4.3	36.7 ± 3.7	36.0 ± 4.0	36.1 ± 4.2	36.4 ± 5.1
P_aw-peak_ ^aaa, ccc^ (cmH_2_0)	ARDSnet ^fff, g^ OLC EIT	21.1 ± 3.5	33.4 ± 3.9	37.5 ± 4.3	37.4 ± 3.5	36.6 ± 2.9	36.1 ± 2.9	40.5 ± 14.2	38.4 ± 5.9
			33.4 ± 3.9	37.5 ± 4.3	37.4 ± 3.5	36.6 ± 3.0	36.1 ± 2.9	41.5 ± 14.2	38.4 ± 5.9
			37.8 ± 8.3	40.3 ± 4.7	41.8 ± 4.8	43.0 ± 4.1	42.5 ± 4.3	42.8 ± 4.7	44.1 ± 6.9
PaO_2_) ^aaa, bbb, ccc, ddd^ (mmHg, 100% 0_2_)	ARDSnet ^eee, fff^ OLC EIT	475 ± 70	129 ± 64	226 ± 100	236 ± 112	192 ± 98	155 ± 64	134 ± 58	184 ± 95
			66 ± 18	442 ± 64	454 ± 79	477 ± 40	471 ± 38	444 ± 42	446 ± 31
			77 ± 32	507 ± 56	502 ± 51	500 ± 40	466 ± 49	430 ± 36	422 ± 74
PaCO_2_ ^aaa^ (mmHg)	ARDSnet OLC EIT	51.4 ± 9.7	58.3 ± 15.0	67.3 ± 21.4	62.4 ± 10.0	58.5 ± 8.5	59.9 ± 9.5	59.0 ± 8.6	62.1 ± 12.1
			60.2 ± 11.9	59.1 ± 9.6	54.7 ± 6.3	57.1 ± 7.2	53.7 ± 6.3	55.9 ± 8.7	56.4 ± 7.7
			58.5 ± 7.9	69.7 ± 10.4	67.8 ± 10.2	68.4 ± 12.4	68.2 ± 16.5	65.1 ± 10.4	67.3 ± 11
Compliance ^aaa, bbb^, (mL·cmH_2_O^−1^) ^ccc, dd^	ARDSnet ^eee, ff^ OLC EIT	26.5 ± 4.7	12.4 ± 2.1	11.6 ± 2.1	11.4 ± 2.4	12.2 ± 2.9	12.0 ± 3.6	11.93.2	11.4 ± 2.3
			13.6 ± 2.	19.8 ± 4.1	19.7 ± 4.5	21.5 ± 7.1	21.9 ± 6.7	21.9 ± 6.1	21.9 ± 5.4
			14.5 ± 4.	18.7 ± 3.8	19.9 ± 4.2	18.9 ± 4.0	18.8 ± 2.7	19.8 ± 3.9	18.9 ± 4.1

Data are shown for the three ventilation groups ARDSnet, OLC, and EIT as described before. Heart rate (HR), mean arterial pressure (MAP), cardiac output (CO), systemic vascular resistance (SVR), pulmonary vascular resistance (PVR), positive end-expiratory pressure (PEEP), driving pressure, plateau airway pressure (P_aw-plat_), peak inspiratory airway pressure (P_aw-peak_), arterial partial pressure of oxygen (PaO_2_, ventilation with 100% oxygen), arterial partial pressure of carbon dioxide (PaCO_2_), and respiratory system compliance (compliance) are given as mean with standard deviation. Because randomized group allocation happened only after induction of ARDS, baseline measurements are averaged over all animals. Superscript letters are used to summarize statistically significant differences; one letter refers to *p* < 0.05, two letters to *p* < 0.01, and three letters to *p* < 0.001. ^a^ indicates statistically significant differences between baseline and ARDS, this was analyzed by paired t-tests over all animals (*n* = 24), irrespective of the group, to test the effects of ARDS induction. ^b^ marks parameters, for which the general linear model (GLM) statistics over all measurements from ARDS until 24 h indicated significant changes over time. ^c^ marks parameters, for which the GLM indicated significant between-group differences and ^d^ significant interaction terms (group * time). ^e^ refers to ARDS vs. OLC, ^f^ to ARDS vs. EIT, and ^g^ to OLC vs. EIT post-hoc tests.

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
