# Peer review of "Mechanical Ventilation Strategies Targeting Different Magnitudes of Collapse and Tidal Recruitment in Porcine Acid Aspiration-Induced Lung Injury"

_jcm, 2019, doi:10.3390/jcm8081250_

Round 1

Reviewer 1 Report

Dear Authors,

I appreciated the manuscript and the experiments were well designed.

There is only some minor methodological issue that needs to be addressed.

First of all, in my opinion the anesthetic drug that you use has to be specified.  In fact, it could affect the hemodynamic parameters. Furthermore, the staining used to obtain the histological images needs to be espicited too.

Author Response

Dear Reviewer, 

thank you very much for your comments. We kindly ask you to see the attachment below. 

J.Haase, C.Buchloh, A. Reske, H.Wrigge

Response to Reviewer 1:

Comment 1:

First of all, in my opinion the anesthetic drug that you use has to be specified.  In fact, it could affect the hemodynamic parameters.

Thank you for your comment. Details on anesthesia were added in the revised manuscript (p.03, lines 118-121):

“Pigs were fasted and premedicated with midazolam (1mg·kg-1intramuscular) and ketamine (15mg·kg-1intramuscular). Animals were anaesthetized by infusion of fentanyl 5(3-30) mg·kg-1·h-1, midazolam 2 (1-6) mg·kg-1·h-1, ketamine 15 (5-30) mg·kg-1·h-1and pancuronium 0.15mg·kg-1·h-1.”

Comment 2:

Furthermore, the staining used to obtain the histological images needs to be espicited too.

We added the following information on p. 5, lines 193-194:

“Tissue samples were fixed per standardized protocol and stained with hematoxylin-eosin (see SM).”

The respective paragraph in the Supplementay Material (SM) reads as follows:

“Afterwards tissue samples were embedded in paraffin at 60°C, cut into 5µm slices and stained with hematoxylin-eosin. Microscopic images from a light microscope (Zeiss Axio Imager with Axio Cam MRc5, Carl Zeiss AG, Feldbach, Switzerland) for analysis were taken in two different magnifications (100x, 400x) from four non-overlapping fields of view per slide.”

Reviewer 2 Report

Journal:       Journal of Clinical Medicine

Manuscript Number: jcm-545416

Title:             Mechanical Ventilation Strategies Targeting Different Magnitudes of Collapse and Tidal Recruitment in Porcine Acid Aspiration Induced Lung Injury

Authors:      Juliane Haase, et al.

1. General Comments

Authors investigated the benefit of positive end-expiratory pressure (PEEP) settings according to the electrical impedance tomography (EIT), compared with the open lung concept (OLC) or the conventional settings recommended by the Acute Respiratory Distress Syndrome network (ARDSnet). They demonstrated that the histological scores of edema and inflammation in ARDSnet were significantly severer compared with the EIT and OLC groups

Because exploring the optimal PEEP settings could be clinically important for improving the prognosis in patients with ARDS, this paper includes novel findings. However, there seem to be several drawbacks prohibiting publication in the current form.

2. Major Comments

1)      Unclear benefit of EIT- and OLC-based PEEP selections: Authors concluded that both EIT- and OLC-based PEEP selections were beneficial for lung-protective mechanical ventilation. However, the pathological examination revealed a significant difference only in the severity of inflammation between the ARDSnet group and the OLC group. Regarding edema and hemorrhage, there were no specific differences among groups, suggesting that the authors’ conclusion seemed too exaggerated.

2)      Debatable protocol of the study: Authors compared EIT- and OLC-based PEEP selections with the ARDSnet protocol with low PEEP strategy. However, the ARDSnet protocol with low PEEP strategy aimed to the permissive atelectasis, whereas the other two strategies aimed to keep airway open. If the authors intended to compare the effects of permissive atelectasis and open lung strategy, this protocol is acceptable, which was consistent with the ART study (JAMA 2017, PMID: 28973363). However, if the authors aimed to compare the different protocols for open lung strategy, the ARDSnet protocol with “high” PEEP strategy should be the control cohort.

3. Minor Comments

3)           The rationale for setting the study period of 24 hours was unclear, which should be clarified. 

Author Response

Dear Reviewer, 

thank you very much for your comments. The answers to your comments can be seen in the attachment.

Sincerely,

J.Haase, C.Buchloh, H.Wrigge and A. Reske 

Response to Reviewer 2:

Major Comment 1:

Unclear benefit of EIT- and OLC-based PEEP selections: Authors concluded that both EIT- and OLC-based PEEP selections were beneficial for lung-protective mechanical ventilation. However, the pathological examination revealed a significant difference only in the severity of inflammation between the ARDSnet group and the OLC group. Regarding edema and hemorrhage, there were no specific differences among groups, suggesting that the authors’ conclusion seemed too exaggerated.

Thank you for this comment, we understand the criticism. We found significant differences in the overall statistical testing of edema and inflammation but not for hemorrhage. Between group differences, however, were only significant for inflammation between ARDSnet group and OLC, while we observed a tendency towards higher scores for ARDSnet compared to OLC and EIT in edema. In line with improved physiological and morphological (CT and EIT) criteria in EIT and OLC group, we interpreted this as potentially less injurious in our model. According to the reviewers comment, we partially rephrased the Abstract (p2, lines 55-58), Discussion (P10, Lines 328-334, P 11 lines 408-409,416-418), Limitations (p12, lines 455-456 and Conclusion (p12, lines 463-466) to clarify this point.

Major Comment 2:     

Debatable protocol of the study: Authors compared EIT- and OLC-based PEEP selections with the ARDSnet protocol with low PEEP strategy. However, the ARDSnet protocol with low PEEP strategy aimed to the permissive atelectasis, whereas the other two strategies aimed to keep airway open. If the authors intended to compare the effects of permissive atelectasis and open lung strategy, this protocol is acceptable, which was consistent with the ART study (JAMA 2017, PMID: 28973363). However, if the authors aimed to compare the different protocols for open lung strategy, the ARDSnet protocol with “high” PEEP strategy should be the control cohort. 

From todays point of view this aspect is absolutely legitimate. When we started the conception of the study ARDSnet low PEEP table was standard of care in the treatment of ARDS patients in many Intensive Care Units (ICUs). Controlled studies and/or metaanalyses indicating that high PEEP treatment [1, 3] with or without recruitment maneuvers [2] may be beneficial in patients with more severe ARDS appeared only after we had designed and registered our study [1-4]. Aim of this prospective experimental study was to compare MV protocols targeting at different magnitudes of collapse and tidal recruitment. We did not aim to compare different open-lung-strategies. Surprisingly, we found comparably high PEEP levels for OLC and EIT guided strategies [4], despite the entirely different approaches to set PEEP. Since the results showed similar levels of PEEP in OLC and EIT group, interpretation of results was conceptionally presented this way in the manuscript.

Minor Comments

The rationale for setting the study period of 24 hours was unclear, which should be clarified

The period was arbitrarly chosen based on experiences from former studies. Based on the intention to conduct a ventilation period longer than acute experimental studies (4-6 hours) and to assess histological changes within the exsudative phase of 72 hours (proteinaceous edema, alveolar hemorrhage and neutrophil infiltration) in early stages of diffuse alveolar damage a period of 24 hours was considered suitable (Cinella et al. 2015).

Furthermore, the German protection of animals act was taken into consideration, which requires to balance a longer study duration (and possible animal suffering) with scientific benefit.

References:

1          Briel M, Meade M, Mercat A, Brower RG, Talmor D, Walter SD et al. Higher vs lower positive end-expiratory pressure in patients with acute lung injury and acute respiratory distress syndrome: systematic review and meta-analysis. JAMA 2010; 303(9):865–73. PMID:20197533, DOI:10.1001/jama.2010.218

2          Goligher EC, Hodgson CL, Adhikari, Neill K J, Meade MO, Wunsch H, Uleryk E et al. Lung Recruitment Maneuvers for Adult Patients with Acute Respiratory Distress Syndrome. A Systematic Review and Meta-Analysis. Ann Am Thorac Soc. 2017; 14(Suppl_4):S304-S311. PMID:29043837, DOI:10.1513/AnnalsATS.201704-340OT

3          Amato MB, Meade MO, Slutsky AS, Brochard L, Costa EL, Schoenfeld DA, Stewart TE, Briel M, Talmor D, Mercat A, Richard JC, Carvalho CR et al. Driving pressure and survival in the acute respiratory distress syndrome. N Engl J Med 2015; 372(8):747-55,  DOI: 10.1056/NEJMsa1410639

4          Franchineau G, Bréchot N, Lebreton G, Hekimian G, Nieszkowska A, Troullet JL, Leprince P, Chastre J, Luyt CE, Combes A, Schmidt M. Bedside Contribution of Electrical Impedance Tomography to Setting Positive End-Expiratory Pressure for Extracorporeal Membrane Oxygenation–treated Patients with Severe Acute Respiratory Distress Syndrome. Am J Respir Crit Care Med 2017; 196(4):447-457. DOI: 10.1164/rccm.201605-1055OC.

Round 2

Reviewer 2 Report

Journal:       Journal of Clinical Medicine

Manuscript Number: jcm-545416.R1

Title:             Mechanical Ventilation Strategies Targeting Different Magnitudes of Collapse and Tidal Recruitment in Porcine Acid Aspiration Induced Lung Injury

Authors:      Juliane Haase, et al.

General Comments

Authors investigated the benefit of positive end-expiratory pressure (PEEP) settings according to the electrical impedance tomography (EIT), compared with the open lung concept (OLC) or the conventional settings recommended by the Acute Respiratory Distress Syndrome network (ARDSnet). They demonstrated that the histological scores of edema and inflammation in ARDSnet were significantly severer compared with the EIT and OLC groups

The authors adequately revised the manuscript, according to the reviewers’ recommendations. This form of paper seems suitable for publication.